# Conservative Surgery in cT4 Breast Cancer: Single-Center Experience in the Neoadjuvant Setting

**DOI:** 10.3390/cancers15092450

**Published:** 2023-04-25

**Authors:** Antonio Franco, Alba Di Leone, Alessandra Fabi, Paolo Belli, Luisa Carbognin, Elisabetta Gambaro, Fabio Marazzi, Elena Jane Mason, Antonino Mulè, Armando Orlandi, Antonella Palazzo, Ida Paris, Alessandro Rossi, Lorenzo Scardina, Daniela Andreina Terribile, Giordana Tiberi, Diana Giannarelli, Giovanni Scambia, Riccardo Masetti, Gianluca Franceschini

**Affiliations:** 1Breast Unit, Department of Women, Children and Public Health Sciences, Fondazione Policlinico Universitario “A. Gemelli” IRCCS, 00168 Roma, Italy; antonio.franco@guest.policlinicogemelli.it (A.F.); alba.dileone@policlinicogemelli.it (A.D.L.);; 2Precision Medicine Senology, Fondazione Policlinico Universitario “A. Gemelli” IRCCS, 00168 Roma, Italy; 3Diagnostic Imaging, Oncological Radiotherapy and Hematology, Fondazione Policlinico Universitario “A. Gemelli” IRCCS, 00168 Roma, Italy; 4Cancer Gynaecology, Department of Women, Children and Public Health Sciences, Fondazione Policlinico Universitario “A. Gemelli” IRCCS, 00168 Roma, Italy; 5Cancer Radiation Therapy, Department of Diagnostic Imaging, Oncological Radiotherapy and Hematology, Fondazione Policlinico Universitario “A. Gemelli” IRCCS, 00168 Roma, Italy; 6Anatomic Pathology, Department of Women, Children and Public Health Sciences, Fondazione Policlinico Universitario “A. Gemelli” IRCCS, 00168 Roma, Italy; 7Medical Oncology, Department of Medical and Surgical Sciences, Fondazione Policlinico Universitario “A. Gemelli” IRCCS, 00168 Roma, Italy; 8Department of Movement, Human and Health Sciences, Università degli Studi di Roma “Foro Italico”, 00135 Roma, Italy; 9Epidemiology and Biostatistics, Fondazione Policlinico Universitario “A. Gemelli” IRCCS, 00168 Roma, Italy; 10Gynecological Oncology Unit, Department of Woman and Child Health and Public Health, Woman Health Area, Fondazione Policlinico Universitario “A. Gemelli” IRCCS, 00168 Roma, Italy

**Keywords:** cT4 breast cancer, neoadjuvant treatment, personalized therapy, inflammatory breast cancer, conservative breast surgery, oncological outcomes

## Abstract

**Simple Summary:**

Screening programs are increasingly leading to a decrease in the diagnosis of locally advanced breast cancer, especially cT4 breast cancer. Currently, therapy is based on a definite scheme consisting of neoadjuvant chemotherapy (NA), surgical therapy, radiation therapy, and possible adjuvant therapy, regardless of the biological histotype. Surgical therapy for this type of treatment has been based on demolitive surgery aimed at achieving complete removal of the neoplasm without consideration of aesthetic outcomes. Recently, conservative surgery has progressively increased in importance in patients with cT4, especially in the presence of a major neoplastic response to chemotherapy. To date, however, few studies have compared the two types of surgery in terms of oncological outcomes (loco-regional disease-free survival, distant disease-free survival, and overall survival). Our aim was to compare these two types of surgery to assess the safety of conservative versus radical therapy.

**Abstract:**

Background: The diffusion of screening programs has resulted in a decrease of cT4 breast cancer diagnosis. The standard care for cT4 was neoadjuvant chemotherapy (NA), surgery, and locoregional or adjuvant systemic therapies. NA allows two outcomes: 1. improve survival rates, and 2. de-escalation of surgery. This de-escalation has allowed the introduction of conservative breast surgery (CBS). We evaluate the possibility of submitting cT4 patients to CBS instead of radical breast surgery (RBS) by assessing the risk of locoregional disease-free survival, (LR-DFS) distant disease-free survival (DDFS), and overall survival (OS). Methods: This monocentric, retrospective study evaluated cT4 patients submitted to NA and surgery between January 2014 and July 2021. The study population included patients undergoing CBS or RBS without immediate reconstruction. Survival curves were obtained using the Kaplan-Meyer method and compared using a Log Rank test. Results: At a follow-up of 43.7 months, LR-DFS was 70% and 75.9%, respectively, in CBS and RBS (*p* = 0.420). DDFS was 67.8% and 29.7%, respectively, (*p* = 0.122). OS was 69.8% and 59.8%, respectively, (*p* = 0.311). Conclusions: In patients with major or complete response to NA, CBS can be considered a safe alternative to RBS in the treatment of cT4a-d stage. In patients with poor response to NA, RBS remained the best surgical choice.

## 1. Introduction

In the last few years, the diffusion of breast screening programs has allowed for an increasingly early diagnosis of breast cancer (BC), resulting in a progressive decrease of diagnosis of locally advanced breast cancers (LABC), especially of T4 clinical stage (cT4) [1]. The cT4 BC is divided into four types according to TNM [2,3]: cT4a (cancer with invasion or fixation to chest wall, ribs, intercostals, or serratus anterior muscles); cT4b (ulceration of the skin and/or ipsilateral satellite skin nodules and/or edema of the skin including peau d’orange skin, that does not meet criteria for defining inflammatory carcinoma); cT4c (simultaneous presence of T4a and T4b characteristics); cT4d or inflammatory carcinoma (IBC) (diffuse erythema and edema, which has a peau d’orange appearance that involves the majority of the breast, early dermal lymphatic and vascular invasion by tumor emboli, rapid tumor growth) [4]. IBC represents about 5% of the BC and it is responsible of 7% of all deaths related to BC [5,6,7,8,9].

The standard care for cT4 BC is a multidisciplinary approach independently from histotype: neoadjuvant treatment, (NA) followed by surgery, adjuvant therapy (chemotherapy, biological therapy, radiotherapy, and eventually hormonal therapy in luminal tumors) [10,11]. Among the advantages of the neoadjuvant approach, there are two important aims: 1. reaching survival rates comparable to early breast cancer, and 2. de-escalation of surgery on breast and axilla. In an increasing number of patients this de-escalation has allowed for conservative breast surgery (CBS) as quadrantectomy (Q) or Level II oncoplastic surgery (OPSII) and conservative mastectomy (Skin sparing mastectomy SSM—or Nipple sparing—NSM—mastectomy). The conversion rate to conservative surgery post NA varies from 28% to 98% of patients [2,3].

Nowadays, there is an unclear consensus as to whether conservative surgery can be performed in patients with cT4a-d stage. Therefore, the treatment of these BC stages is based almost exclusively on radical breast surgery (RBS) and in a smaller number of cases on surgery conservation [2,3,12,13]. CBS is recommended especially in patients responsive to NA. RBS is still indicated in patients resistant to NA or in cIBC (cT4d) [14]. Few researchers have explored the efficacy of CBS in patients with cT4 stage submitted to NA.

The purpose of this study is to evaluate the possibility of treating patients with locally advanced breast cancer (cT4a-d), after initially undergoing neoadjuvant chemotherapy, by subsequently using conservative surgery instead of radical surgery while preserving the same oncologic outcomes: Locoregional disease-free survival (LR-DFS); Distant disease-free survival (DDFS); Overall survival (OS).

## 2. Materials and Methods

### 2.1. Patients Characteristics

This is a retrospective, observational, monocentric study conducted at Policlinico Universitario A. Gemelli IRCCS, Rome. This study evaluated patients affected by locally advanced BC cT4 with indication to NA therapy and subsequent surgery in accordance with national guidelines and internal management [2,3], from January 2014 to July 2021 and followed until 31 December 2021. The study population included consecutive patients with cT4 breast cancer studied with complete diagnostic imaging and undergoing CBS (Quadrantectomy with or without level II oncoplastic surgery—OPSII—and conservative mastectomy with immediate reconstruction) or RBS without immediate reconstruction (modified radical mastectomy—MRM). These two groups were identified with the aim of assessing whether preservation of the mammary gland together with the skin exposes patients to a higher risk of recurrence than surgery involving complete removal of the mammary gland and skin covering. All biological subtypes were included in the analysis, while those excluded were cT1-3 BCs as well as those with previous or synchronous history of systemic malignant neoplasms or stage IV patients or evidence of metastatic widespreading in course of NA.

Data collection from patient records has been updated in a database managed by the Breast Center Unit. A multidisciplinary team composed of breast and plastic surgeons, oncologists, radiotherapists, radiologists, pathologists, geriatrics, psychologists, geneticists, and a case manager planned each patient’s treatment. Patients were followed by clinicians and evaluated as outpatients [15]. The clinical stage (TNM and extension, multifocality and skin or muscle involvement) was defined by clinical evaluation, mammography, breast and axillary echotomography, and magnetic resonance imaging (MRI). A total body TC scan was performed before the start of systemic treatment. Patients with a cT4 stage and any nodal status (cN0-3) were submitted to NA treatment.

A number of clinical, pathological, and biological features were considered for analysis: site of neoplasm; size assessed by MRI; tumor istotype (Ductal invasive Carcinoma—DIC; Lobular Invasive Carcinoma—LIC; Invasive Carcinoma No Special Type—IC NST); grade (G 1,2,3); expression of estrogen and/or progesteron receptors; proliferation index (Ki67); presence of HER2; neoplastic subtypes as luminal, Triple Negative (TN) and HER2 positive (HER2+). A metal trace (amagnetic clip) was placed in the neoplasm in all patients.

In the post NA treatment, for the pre-operative re-stadiation, echotomography, mammography, and MRI were used to assess tumor response to chemotherapy and classified according to RECIST1.1 [16,17]. In addition, we also evaluated the reduction of the maximum tumor size at MRI expressed as a percentage of reduction. Considering the response, we identified three classes: complete reduction (absence of tumor—100%); major reduction (99.9–25.1%); minimal reduction or progression of disease (<25%).

### 2.2. Neoadjuvant Treatment

The NA treatment consisted mainly of anthracyclines and taxanes including regimens lasting for 6 months. In HER2+ patients, three-weekly Trastuzumab was added for a total of 1-year administration [18]. In patients with triple negative cancer, carboplatin could be added to Paclitaxel.

### 2.3. Surgery and Pathological Evaluation

The type of surgery was defined by a multidisciplinary team. The type of CBS was decided on the basis of different features:Skin involvement post NA. In case of extensive skin involvement, patients underwent RBS without reconstruction; in case of limited involvement or complete resolution, patients underwent conservative surgery by removing the affected skin area.Extension of muscle involvement post NA. Patients with extensive muscle involvement underwent RBS, while for those with limited involvement, conservative surgery was considered by removing the involved muscle area.Extension of neoplastic residue at radiological stadiation post NA. In the presence of <20% involvement of the mammary gland there was indication for Q, between 20% and 50% quadrantectomy with OPSII and >50% conservative mastectomy with immediate reconstruction [19].Site of neoplasm and expected aesthetic outcome. Lesions of the upper-outer quadrant (UOQ) or axillary tail (AT) or external quadrant had indication to quadrantectomy (Q), lesions of the inner quadrants (as upper-inner quadrant—UIQ or sub-areolar quadrant-SQ—had indication to OPSII and finally lesions of the lower-inner quadrant (LIQ) had indication to conservative mastectomy.Presence of any pathogenic mutations that were subjected to conservative mastectomy [20].Patient’s preferences.

During the surgical procedure we planned to remove the amagnetic clip, thus allowing precise identification of the surgical bed even in case of complete radiological response. In all patients the surgical bed was excised.

The axillary surgical approach was related to NA clinical response. Patients achieving clinical negativity of lymph nodes or clinical complete response (absence of clinically and radiologically tumor [cN0 − ycN0/cN + ycN0] underwent sentinel lymph-node biopsy (SLNB) with histological evaluation. Patients with macro- or micrometastases evidence in histopathological evaluation underwent axillary dissection (AD). Patients with clinically or radiologically evidence of lymph-nodes post NA (ycN+) were submitted directly to AD without SLNB. During surgery, all patients went through systematic shaving of surgical bed tumor in order to evaluate any neoplastic infiltration adjacent to the operative piece [21]. In all types of surgery, the infiltrated skin or muscle after NA was removed. Whereas in cases of cT4b with complete response the skin was preserved.

For pathological evaluation, we used the following criteria: 1. infiltration of the skin or muscle excision; 2. infiltration of the margins by the invasive or in situ carcinoma (DCIS); 3. presence of both invasive and DCIS on re-resected margins. The complete pathological response (pCR) was defined as the absence of invasive tumor [22]. The presence of DCIS was considered for evaluating the “ink on tumor” and close margins, in accordance with guidelines [2,3]. Patients underwent a new surgery if infiltrations of margins were documented.

### 2.4. Adjuvant Treatment

The use of adjuvant therapy was confirmed in relation to the type of surgery, the tumor biology, and definitive pathological stadiation. In case of residual disease, patients with TN subtype received in most cases capecitabine [23,24,25], patients with HER2+ tumor were given Trastuzumab or Trastuzumab Emtansine (TDM-1) for up to 14 cycles, in relation to time of cancer diagnosis [26]. Patients with estrogen receptor/progesteron receptor (ER/PGR) expression ≥1% received endocrine therapy (Tamoxifen or Aromatase inhibitors with or without LHRH analogue, if in a pre- or postmenopausal status respectively) for a maximum of 7–10 years [27,28]. Radiation therapy was conducted in accordance with the applicable international guidelines [29,30].

### 2.5. Follow-Up and Oncological Outcomes

All patients were evaluated during follow-up by outpatient visit or telephone interview, especially during the SARS-CoV-19 pandemic, and were followed every six months for the first three years, then every twelve months [31]. Follow-up included locoregional assessment performed by breast and axillary ultrasound every six months and mammography every twelve months. In the presence of breast prosthesis, annual MRI was also added. Systemic stadiation was obtained in accordance with guidelines.

The group of patients undergoing CBS was compared with those who underwent RBS by assessing the risk of locoregional recurrence (LRDFS), risk of systemic recurrence (DDFS), or overall survival (OS) considered as the time (months) between the first date of chemotherapy and any event of death or the last known follow-up.

### 2.6. Statistical Analysis

Continuous variables were described by mean ± standard deviation (SD) (median and interquartile range) and compared in the subgroups with Student’s t test. Categorical variables have been described by absolute number and percentage and associations among them were assessed with the chi-square test. Univariate and multivariable analyses were conducted using binary logistic regression and aimed to identify predisposing factors to CBS. Odds Ratios (OR) were reported along with their 95% confidence intervals. Survival curves were obtained using the Kaplan-Meier method and compared by the Log Rank test. All statistical evaluations were two-tailed and considered significant if *p*-value < 0.05 (*p* < 0.05). Statistical analysis was performed using SPSS ver. 26.0 (Statistical Package of Social Science).

## 3. Results

Among 1089 patients who underwent NA treatment, a total of 129 (11.8%) cT4 consecutive patients were observed. Thirty-two patients were excluded from the analysis: thirteen (10.1%) for BC metastatic disease at diagnosis, eleven (8.5%) due to endocrine therapy as neoadjuvant therapy, four (3.1%) for a previous diagnosis of BC, three (2.3%) for developing metastasis during NA, and two (1.6%) patients for a previous history of malignant neoplasms (pancreatic and laryngeal cancer). The remaining 96 (74.4%) patients were entered into the study analysis. Table 1 shows the patient characteristics.

The mean age was 53.7 ± 11.3 (52; 45.5–62.3). Fifty-seven (59.4%) patients presented menopausal status. Six (6.3%) showed BRCA 1/2 pathological mutations. Body mass index (BMI) was 25.8 ± 5.1 (25; 22.1–28.6) Kg/m^2^. No differences were found between the two groups concerning epidemiological characteristics. In more than half of the patients, the site of the tumor was the upper-outer quadrant/axillary tail (UOQ/AT) (57–61.4%).

At diagnosis, only one patient (1%) showed pectoral muscle involvement (4a). Fifty-eight (60.4%) patients showed skin involvement (4b). Nine (9.4%) presented an involvement of both skin and muscle (4c). Twenty-eight (29.2%) presented inflammatory BC (4d). The most prevalent histotype was IDC (54 patients—56.3%) with grade three (59 patients—61.5%). Luminal A, Luminal B, HER2 positive, and TN biological subtypes were presented in seven (7.3%), forty-one (42.7%), twenty-nine (30.2%), and nineteen (19.8%) patients, respectively (*p* = 0.900). In the CBS group, no significative difference was shown in terms of tumor size, biological isotype, grading, and tumor subtype. The only statistically significant difference between the two groups were cT4 type and stage at diagnosis. IBC (cT4d) was more frequent in the RBS group (46.7% vs. 13.7%) than non-inflammatory carcinoma (cT4a-c) (53.3% vs. 86.3%) (*p <* 0.0001) and the more advanced stage (IIIC) underwent much more RBS (24.4% vs. 5.9%—*p* = 0.018).

Radiological features were described by MRI (Table 2).

The largest tumor size was 57.4 ± 26.7 (51.5; 39.3–73.8). Considering the conservative and radical surgery groups, patients with larger neoplasm size underwent RBS (*p* = 0.046). The same trend was confirmed considering the size of the residual neoplasm; patients with more residual tumor post NA underwent RBS (39.6 ± 32.8 vs. 21.9 ± 20.4—*p* = 0.002). No difference was found between the two groups regarding axillary involvement at diagnosis (cN). Patients with stability of disease or progression were more likely to undergo RBS (44.4% vs. 19.6%—*p* = 0.036).

Considering the RECIST criteria, we identify no significant difference between the groups undergoing conservative and radical surgery. Instead, considering the percentage of cancer reduction, twenty patients (20.8%) reported a complete radiological response, forty-six patients (47.9%) reported a partial response (>25%), and thirty patients (31.3%) showed stability or local progression.

Table 3 summarizes the type of surgery after NA.

All patients undergoing RBS received a modified radical mastectomy (MRM) without reconstruction (45–100%). The majority of those undergoing conservative surgery received a quadrantectomy (25–49.0%). Eight patients (15.8%) underwent Level II Oncoplastic surgery and eighteen (35.2%) had the conservative mastectomy with immediate reconstruction. In all cases of conservative or radical surgery, the skin and/or muscle area involved by the neoplasm was removed.

### 3.1. Pathological Results

In Table 4 pathological characteristics after surgery are shown.

A total of twenty-five (26.0%) patients achieved a pathological complete response. No significant difference was found between the two groups regarding the percentage of pCR (31.4 vs. 20%—*p* = 0.248). As noted in the post NA assessment, also in pathologic evaluation all patients showing skin (thirteen cases) and/or muscle (five cases) involvement received RBS. While patients with smaller residual lesion underwent more frequently CBS (49.0% CBS vs. 20.0% RBS—*p* < 0.0001). This trend was also confirmed by the pathological status assessment. In fact, patients with major or complete responses, (pathological status 0, I, and IIA had more frequently received a CBS than patients who showed more advanced stages (IIB; IIIA; IIIB and IIIC). RBS also showed more positive lymph nodes for neoplastic cells (6.1 vs. 2; *p = 0.001*) and a worse positive lymph nodes/total lymph nodes ratio (0.41 vs. 0.15—*p <* 0.00001), respectively.

Table 5 underlines the univariate and multivariable analysis of the features predisposing to conservative surgery.

We found among the factors predisposing to conservative surgery highlighted in the univariate analysis: the presence of cT4b tumor (OR 6.530—*p* < 0.0001) and a tumor size less than 50 mm at diagnosis (OR 2.491—*p* = 0.033). These were the predictive factors of RBS: cT4d (OR 0.168—*p* = 0.001); tumor size greater than 80 mm (OR 0.318—*p* = 0.026); tumor reduction < 25% or progression (OR 0.305—*p* = 0.010); and presence of complete clinical response on N (ycN0) (OR 0.331—*p* = 0.009).

Finally, in the multivariable analysis, we found two predictive factors of conservative surgery: cT4b at diagnosis as favorable factors (OR 6.941—*p* = 0.021; 95% CI 1.245–35.543) and tumor reduction < 25% or progression during NA treatment as unfavorable factors (OR 0.291—*p* = 0.042; 95% CI 0.090–0.955).

### 3.2. Clinical Outcome

At a follow-up of 43.7 ± 21.9 months (43.2; 25.7–60.9), nineteen (19.8%) patients presented locoregional relapse (eleven and eight breast and axillary recurrence, respectively), while thirty-four patients (35.4%) presented distant recurrence. Twenty-nine (30.2%) patients died due to disease systemic progression (Table 6).

Considering the response to NA therapy, most of the relapses were shown in patients achieving progression or lower response following NA. Out of fifty-one patients who underwent CBS, eleven (21.6%), eleven (21.6%), and nine (17.6%) experienced loco-regional recurrence, distance recurrence, and death, respectively. In patients who underwent RBS, eight (17.8%), twenty-three (51.5%), and twenty (44.4%) showed locoregional (LR) recurrence, distance relapse, and death, respectively. CBS showed a better DDFS (64.8% vs. 44.7%—*p* = 0.002) and overall survival (OS) (71.2% vs. 48.6%—*p* = 0.005) compared to RBS. No difference was shown between LR-DFS (*p* = 0.798) (Figure 1).

By dividing patients on the basis of radiological response to chemotherapy, the group of patients with complete reduction (100%) and major reduction (99.9–25.1%) to NA treatment consisted of sixty-six patients. Among them, forty (60.6%) patients had no events at the end of the analysis periods. The remaining patients (26–39.4%) reported at least one occurrence: fourteen (21.2%) showed locoregional recurrence, nineteen (28.8%) systemic recurrence, and fourteen (21.2%) died. While the group of patients with minimal reduction or progression of disease (<25%) consisted of thirty patients. Fifteen patients (50%) reported no event, while fifteen (50%) reported at least one occurrence: five (16.7%) showed locoregional recurrence, fifteen (50%) systemic recurrence, and thirteen (43.3%) died.

The two groups manifested a significant difference in survival regarding DDFS (57.4–47.6%—*p* = 0.009) and OS (66.1–49.7%—*p* = 0.008) (Figure 2).

In patients with minor response or progression during NACT, the surgery has too little importance from a curative point of view, but only one of local sanitation, hemostatic, cytoreductive, and this time also cosmetic.

Therefore, considering the sixty-six patients with greater or complete response to NA, locoregional DFS was not statistically different between the two groups. Overall LR-DFS for the CBS group was 70%, whereas for the RBS group it was 75.9% (*p* = 0.420) (Figure 3A). The DDFS was also comparable in the two groups: overall, 67.8% for CBS and 29.7% for RBS (*p* = 0.122) (Figure 3B). The OS of these patients was not statistically significant between CBS (69.8%) and RBS (59.8%) (*p* = 0.311) (Figure 3C).

The use of quadrantectomy alone, compared to OPSII and conservative mastectomy, does not determine change on LR-DFS (*p* = 0.190), D-DFS (*p* = 0.343), and OS (*p* = 0.493).

One point can be made concerning IBC. CBS was indicated in seven of twenty-eight patients (25%). These patients reported one case of locoregional recurrence and no cases of systemic recurrence and death. While the remaining twenty-one cases undergoing RBS showed four cases of locoregional recurrence, thirteen cases of systemic recurrence, and twelve cases of cancer-related death. Like the remaining population, these data seem to show how CBS would appear to show better oncological outcomes. However, looking only at patients with greater response to therapy, we can see that CBS has similar outcomes to RBS. If we consider only patients with major response to NACT (seventeen cases with five undergoing CBS and twelve undergoing RBS), LR-DFS was 83.3% for CBS and 80% for RBS (*p* = 873); DDFS was 100% vs. 58.3% (*p* = 0.070) and OS was 100% vs. 65.6% (*p* = 0.126), respectively.

## 4. Discussion

With the intensification in screening programs, there has been a gradual decrease in the diagnosis of locally advanced BCs in recent years, especially in cT4 BC. These were currently found to have an incidence of 5% of all BCs and it is responsible for 7% of BC-related mortality [10,32].

In addition, in recent years there has been a change in the treatment of these cancers. NA has become the initial treatment for these patients regardless of the initial histotype. Less certainties exist regarding the type of surgical treatment after NA therapies. Radical surgical treatment remains the surgical choice in patients initially diagnosed with cT4 [33,34]. However, currently, with the introduction of personalized therapies, surgery is increasingly influenced by the response to NA. Few studies, nevertheless, have evaluated the possibility of conservative surgery in those patients.

Recent evidence points out that like locally advanced disease, even in cT4, response to chemotherapy could influence oncological outcomes even in patients undergoing conservative surgery. In our center, all BC patients with an initial diagnosis of cT4 underwent NA. Subsequently, surgery was decided by a multidisciplinary team (MDT) and influenced by response to NA, patient’s age, final cosmetic outcome, and patient’s preferences [8,35]. All those patients who showed stability or local progression during NA underwent RBS. In the remaining cases, surgery was determined on the basis of several parameters including size of residue, site of cancer, skin or muscle involvement, preferred aesthetic outcome, and patient preference [36]. Radical surgery is increasingly giving way to conservative surgery, because it allows for better aesthetic and functional outcomes with less postoperative pain or less incidence of lymphedema. We define RBS as modified radical mastectomy, which is surgery that involves the removal of the entire mammary glandular all together with the skin and nipple areola complex without immediate prosthetic reconstruction. Conservative surgery instead consists of surgery in which only a part of the mammary gland is removed (quadrantectomy or Level II oncoplastic surgery) or in case the entire gland needs to be removed, consists of breast skin preservation with or without areola-nipple complex and immediate prosthetic reconstruction (Nipple or Skin sparing mastectomy—NSM vs. SSM) [37]. Therefore, no evidence points out if these benefits could be provided in cT4 patients without changing oncologic outcomes. Then we reviewed our case series to assess whether conservative surgery has the same oncologic outcomes as radical surgery in patients with cT4 breast cancer at diagnosis.

Ninety-six patients met the inclusion criteria and were included in the analysis. Among them, fifty-one (53.1%) underwent CBS and forty-five underwent RBS (46.9%). In the CBS group, twenty-five underwent Quadrantectomy, eight underwent OPSII, and eighteen underwent conservative mastectomies (nine Nipple-Sparing and nine Skin-Sparing). All patients who underwent RBS received modified radical mastectomy surgeries without reconstruction. The two groups showed similar biological characteristics and tumor subtype. As reported in the literature [12,13,14], in our case series, the presence of inflammatory carcinoma (cT4d) was more frequently subjected to RBS (46.7% vs. 13.7%—*p* < 0.001), while 70.7% of patients with cT4b underwent conservative surgery, with the only indication to remove the skin or muscle area involved by cancer. The only case of cT4a also received conservative surgery, removing the muscle’s area involved by the neoplasm that resulted without cancer residue for complete pathological response.

Three additional differences that influenced the type of surgery were neoplastic size, stage at diagnosis and residual tumor at post NA locoregional stadiation exams. Specifically, in the presence of larger initial tumor size (63.1 ± 26.1 vs. 52.3 ± 26.4—*p* = 0.046) or in the case of larger neoplastic residue after treatment (39.6 ± 32.8 vs. 21.9 ± 20.4—*p* = 0.002), the treatment of choice was radical surgery. Conservative surgery is impossible in the case of larger tumors involving a larger skin or muscle area, due to aesthetics results or because of a tumor retaining its larger size even after NA. Advanced pathological state (stage IIIC) most frequently underwent radical surgery.

Finally, the surgical choice was influenced by the response to chemotherapy to the biological subtype. All tumors that remained or progressed to cT4 (ycT4) during therapy received radical surgery (14 patients—14.6%). Patients who responded mostly or entirely to therapy (13 of 20–65%—among rCR and 12 of 19–63.2%—among ycT1a-c) received conservative surgery.

To further test the influence of response to NA treatment on the type of surgery, we evaluated the percentage of reduction in the maximum diameter of the largest lesion assessed by MRI. We introduced three classes: complete reduction of diameter (100%); major reduction of diameter (from 99.9 to 25.1%); minimal reduction or progression of neoplasm (<25% or progression).

Among twenty patients (20.8%) with evidence of 100% reduction in the maximum size of the main lesion, thirteen underwent conservative surgery (65% of cases). The additional seven cases received radical surgery because three had a post NA stage cT4a and cT4d, two because of patient’s preference, one had bilateral neoplasm that made conservative surgery aesthetically difficult, and one because of massive skin involvement at diagnosis. In contrast, 66.7% patients (twenty of thirty) with less than 25% reduction or increase in maximum lesion size received RBS. Evaluation of the reduction in the maximum size of the main lesion provides a valuable and especially immediate tool in assessing response to therapy that can be used by the surgeon in deciding surgical treatment, thus presenting an effective alternative to RECIST criteria that still require a radiologic evaluation and which are therefore not always available. The assessment in the reduction of the diameter of the maximum dimension was found to be predictive of conservative surgery also in the multivariable analysis (*p* = 0.050), thus resulting in a more effective tool in the pre-surgical evaluation of patients with cT4 BC and especially cT4b patients (*p* = 0.022).

Persistence of axillary neoplasm after NA (ycN+) also influenced the surgical choice versus radical surgery (28 of 46 patients—60.9%). This is because the presence of positive lymph nodes is associated with poor response to neoadjuvant therapy [38].

Concerns for the choice of surgery type were confirmed by the post-surgical pathological assessment. Indeed, all patients who were tested as ycT4 continued to show skin (thirteen cases) or muscle (five cases) involvement (ypT4b-c) and all underwent radical RBS. Those with complete (ypT0) or greater (ypT1a-c) reduction were more prevalent among conservative surgery (41 of 59–69.5%). Further confirmation that response to chemotherapy influences surgical choice is also provided by the evaluation of the post-surgery pathological stage. Indeed, lower stages (tumor responders to NACT) resulted from conservative surgery, while higher stages (less responsive to NACT) resulted from radical surgery.

Similarly, more patients undergoing RBS showed lymph node positivity (ypN+) (23 out of 32–71.8%), a greater number of positive lymph nodes (6.1 vs. 2—*p* = 0.001), and a worse ratio between positive lymph nodes and lymph nodes removed (0.41 vs. 0.15—*p* < 0.0001). Thus, in our case series, radical breast surgery is confirmed in patients unresponsive or poorly responsive to NA in both breast and axillary [39,40].

Therefore, as also confirmed by the univariate analysis, the factors which resulted in influencing the type of surgery towards a more conservative surgery were only the presence of an initial cutaneous involvement (cT4b—*p <* 0.0001) and a smaller lesion (<50 mm—*p* = 0.033). Factors predictive of RBS were found to be the presence of an inflammatory carcinoma (cT4d—*p* = 0.001), a larger neoplastic lesion (>80 mm—*p* = 0.026), and a lower response to NA therapies: less than 25% decrease in diameter of the major lesion or progression (*p* = 0.010) and axillary non-response (ycN+—*p* = 0.009).

Regarding the oncological outcomes, patients undergoing CBS seemed to show better oncological outcomes in terms of DDFS (64.8% vs. 44.7%—*p* = 0.002) and OS (71.2% vs. 48.6%—*p* = 0.005). Patients who had a negative outcome after NA inevitably underwent RBS and this could have made the observed difference. Thus, higher incidence of locoregional and systemic recurrence cannot be attributed to more aggressive neoplastic histotypes (as the two groups showed an equal distribution of tumoral subtype) or to the presence of inflammatory carcinoma in the RBS group [40]. This difference could be associated with the fact that all patients who fail to respond to chemotherapy tend to have both RBS and worse systemic outcomes [41]. The lack of pathological complete response represents a negative prognostic factor in terms of disease outcome for patients affected by locally advanced BC [42].

To reduce these biases, we have studied the outcomes taking into account response to NA and patients showing greater response (>25% reduction from initial extent of neoplasm or complete response to NA) and patients showing lesser response (<25% or progression in the maximum diameter of neoplastic lesion). Patients with lower response to chemotherapy or progression showed worse DDFS (57.3% vs. 47.6%—*p* = 0.009) and worse OS (66% vs. 49.7%—*p* = 0.009). These results confirm the worse oncologic outcomes in the totality of patients attributed to RBS [6].

As reported in the literature [43], considering only those patients with a good response to NA, there is no significant difference between CBS and RBS in terms of LR-DFS (70% vs. 75.9%—*p* = 0.420), DDFS (67.8% vs. 29.7%—*p* = 0.112), and OS (69.8% vs. 59.8%—*p* = 0.311). Therefore, RBS intervention remains indicated in patients with poor or no response or progression during NA because they have a high risk of distant recurrence and mortality. In patients sensible to NA, we can provide CBS as it does not alter distant oncologic outcomes. Moreover, in order to further personalize surgery, we can offer the type of intervention that best suits the characteristics of disease with attention to removing the full area of skin or muscle affected by the neoplasm.

A last consideration can be made regarding IBC cases. CBS can also be a valid surgical option in IBCs especially in cases with greater or complete response to NACT. Our cases, although limited in number, showed that oncologic outcomes are similar in these two kinds of surgery only in patients responding to NACT. In those with poor response or progression, the best treatment choice remains RBS [44].

The strength of this study is surely the possibility of evaluating the tumor response simply by imaging available at the time of the decision on the type of surgery. Moreover, cT4 is more infrequent among BCs but is often associated with a difficult surgery that alters the perception of the female body. Therefore, having a prediction of which surgery to perform in the light of clinical and biological characteristics could be important for the true benefit of the patient. The main limitations of this study are the retrospective nature of it and its monocentric nature. A prospective multicenter study is required to confirm these results.

## 5. Conclusions

CBS can be used in patients with initially diagnosed cT4 BC, especially in patients who respond to neoadjuvant chemotherapy. In these patients, CBS showed the same oncologic outcomes as RBS. RBS remains indicated in those patients who respond poorly or who progress in the course of NA.

## Figures and Tables

**Figure 1 cancers-15-02450-f001:**
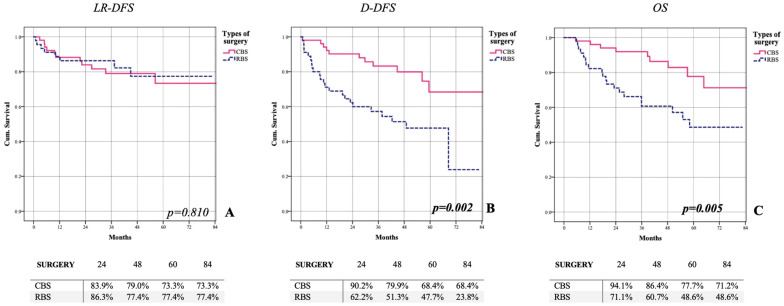
Assessment of oncological outcomes of all patients. Patients who underwent RBS appear to have worse outcomes than patients who underwent CBS concerning locoregional recurrence (**A**); Distant recurrence (**B**) ant overall survival (**C**). (LR-DFS: locoregional disease free-survival; DDFS: distant disease free-survival; OS: overall survival).

**Figure 2 cancers-15-02450-f002:**
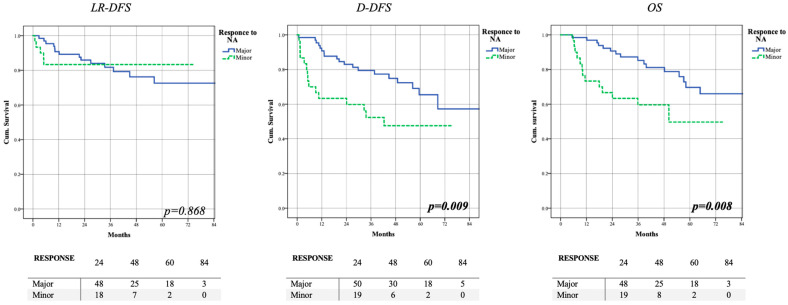
Assessment of oncological outcomes based on response to NACT. Patients with lower response to NA chemotherapy showed worse systemic outcomes. (LR-DFS: locoregional disease free-survival; DDFS: distant disease free-survival; OS: overall survival).

**Figure 3 cancers-15-02450-f003:**
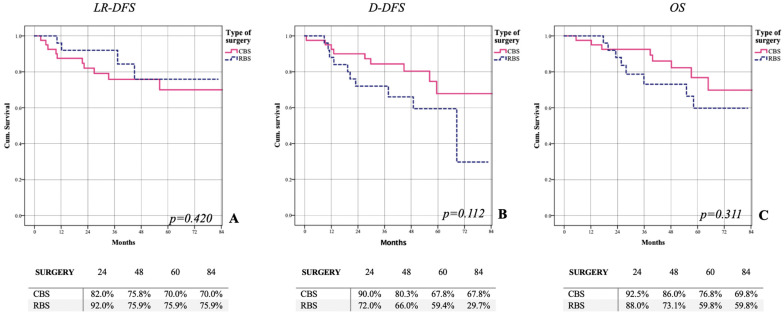
Assessment of oncological outcomes based on type of surgery in patients with major or complete response to NACT. Use of CBS in patients with major response to chemotherapy does not affect LR-DFS (**A**). CBS shows the same D-DFS (**B**) and OS (**C**) than RBS. (LR-DFS: locoregional disease free-survival; DDFS: distant disease free-survival; OS: overall survival).

**Table 1 cancers-15-02450-t001:** Epidemiological, anatomical, and biological features of patients enrolled.

Characteristics	ALL 96 Patients	CBS 51 (53.1%)	RBS 45 (46.9%)	*p*-Value
**Epidemic and anatomical characteristics**				
**Age (years)**	53.7 ± 11.3 (52; 45.5–62.3)	53 ± 11.6 (51; 45.6–60.6)	54 ± 11.4 (53.9; 45.4–63.7)	*p = 0.697*
**Menopausal Status**	57 (59.4%)	29 (56.9%)	28 (62.2%)	*p = 0.679*
**BMI (Kg/m^2^)**	25.8 ± 5.1 (25; 22.1–28.6)	26.1 ± 5 (25.4; 22.3–29.4)	25.5 ± 5.2 (23.8; 22–27.9)	*p = 0.531*
**BRCA 1/2 pathological mutations**	6 (6.3%)	5 (9.8%)	1 (2.2%)	*p = 0.209*
**Tumor site**				*p = 0.159*
**UOQ/AT**	59 (61.5%)	36 (70.6%)	23 (51.1%)	
**UIQ**	3 (3.1%)	1 (2.0%)	2 (4.4%)	
**LIQ**	9 (9.4%)	5 (9.8%)	4 (8.9%)	
**LOQ**	10 (10.4%)	5 (9.8%)	5 (11.1%)	
**SQ**	15 (15.6%)	4 (7.8%)	11 (24.4%)	
**cT4 Types**				** *p < 0.0001* **
- 4a	1 (1.0%)	1 (2.0%)	0 (0%)	
- 4b	58 (60.4%)	41 (80.4%)	17 (37.8%)	
- 4c	9 (9.4%)	2 (3.9%)	7 (15.6%)	
- 4d	28 (29.2%)	7 (13.7%)	21 (46.7%)	
**Initial stage**				***p* = 0.018**
- III B	82 (85.4%)	48 (94.1%)	34 (75.6%)	
- III C	14 (14.6%)	3 (5.9%)	11 (24.4%)	
**Biological characteristics**				
**Istotype**				*p = 0.419*
- DIC	54 (56.3%)	31 (60.8%)	23 (51.1%)	
- LIC	9 (9.4%)	3 (5.9%)	6 (13.3%)	
- IC NST	33 (34.4%)	17 (33.3%)	16 (35.6%)	
**Grading**				*p = 0.675*
- G2	37 (38.5%)	21 (41.2%)	16 (35.6%)	
- G3	59 (61.5%)	30 (58.8%)	29 (64.4%)	
**Estrogen Receptors**				*p = 1.000*
- Negative	32 (33.3%)	17 (33.3%)	15 (33.3%)	
- Positive	64 (56.3%)	34 (66.7%)	30 (66.7%)	
**Progesterone Receptors**				*p = 0.396*
- Negative	35 (36.5%)	21 (41.2%)	14 (31.1%)	
- Positive	61 (63.5%)	30 (58.8%)	31 (68.9%)	
**Ki-67**				*p = 0.468*
- <20	8 (8.3%)	3 (5.9%)	5 (11.1%)	
- ≥20	88 (91.7%)	48 (94.1%)	40 (88.9%)	
**HER2**				*p = 0.723*
- 0	23 (24.0%)	13 (25.5%)	10 (22.2%)	
- 1+	25 (26.0%)	15 (29.4%)	10 (22.2%)	
- 2+ SISH negative	19 (19.8%)	7 (13.7%)	12 (26.7%	
- 2+ SISH positive	8 (8.3%)	5 (9.8%)	3 (6.7%)	
- 3+	21 (21.9%)	11 (21.6%)	10 (22.2%)	
**Tumor subtype**				*p = 0.900*
- Luminal A	7 (7.3%)	4 (7.8%)	3 (6.7%)	
- Luminal B	41 (42.7%)	20 (39.2%)	21 (46.7%)	
- Her2 Positive	29 (30.2%)	16 (31.4%)	13 (28.9%)	
- Triple Negative	19 (19.8%)	11 (21.6%)	8 (17.8%)	

CBS = conservative breast surgery; RBS = radical breast surgery; UOQ = upper-outer quadrant; AT = axillary tail; UIQ = upper internal quadrant; LIQ = Lower internal quadrant; LOQ = lower-outer quadrant; SQ = subarreolar quadrant; DIC = Ductal invasive Carcinoma; LIC = Lobular Invasive Carcinoma; IC-NAS = Invasive Carcinoma No Special Type.

**Table 2 cancers-15-02450-t002:** Radiological assessment pre- and post NA treatment with radiological and clinical evaluation of tumoral response.

Characteristics	ALL 96 Patients	CBS 51 (53.1%)	RBS 45 (46.9%)	*p*-Value
**B** **aseline**				
**Median tumor size (mm)**	57.4 ± 26.7 (51.5; 39.3–73.8)	52.3 ± 26.4 (46; 33–70)	63.1 ± 26.1 (60; 44.5–82.5)	** *p = 0.046* **
**Multifocality**	39 (40.6%)	23 (45.1%)	16 (35.6%)	*p = 0.411*
**Clinical Node Stadiation**				*p = 0.191*
- N0	16 (16.7%)	10 (19.6%)	6 (13.3%)	
- N1	39 (40.6%)	23 (45.1%)	16 (35.6%)	
- N2	27 (28.1%)	15 (29.4%)	12 (26.7%)	
- N3	14 (14.6%)	3 (5.9%)	11 (24.4%)	
**Post-NA radiological characteristics**				
**Residual tumor**	30.2 ± 28.2	21.9 ± 20.4	39.6 ± 32.8	** *p = 0.002* **
** *dimension (mm)* **	(25; 6.3–43.8)	(22; 0–35)	(36; 9–62)	
**ycT**				** *p < 0.00001* **
- **rCR**	20 (20.8%)	13 (25.5%)	7 (15.5%)	
- **1**	19 (19.8%)	12 (23.5%)	7 (15.5%)	
- **2**	32 (33.3%)	23 (45.1%)	9 (20.1%)	
- **3**	11 (11.5%)	3 (5.9%)	8 (17.8%)	
- **4**	14 (14.6%)	0 (0%)	14 (31.1%)	
**RECIST CRITERIA**				*p = 0.097*
- CR	20 (20.8%)	13 (25.5%)	7 (15.6%)	
- PR	41 (42.7%)	24 (47.0%)	17 (37.8%)	
- SD	28 (29.2%)	14 (27.5%)	14 (31.1%)	
- PD	7 (7.3%)	0 (0%)	7 (15.5%)	
**Reduction of maximum size (%)**				** *p = 0.036* **
- 100	20 (20.8%)	13 (25.5%)	7 (15.6%)	
- 99.9–25.1	46 (47.9%)	28 (54.9%)	18 (40.0%)	
- <25 or progression	30 (31.3%)	10 (19.6%)	20 (44.4%)	
**Multifocality**	26 (27.1%)	13 (25.5%)	13 (28.9%)	*p = 0.819*
**ycN**				** *p = 0.007* **
- ycN0	50 (52.1%)	33 (64.7%)	17 (37.8%)	
- ycN+	46 (47.9%)	18 (35.3%)	28 (62.2%)	

CBS = conservative breast surgery; RBS = radical breast surgery; ycT = clinical assessment post NA; rCR = radiological complete response; CR = complete response; PR = partial response; SD = stable disease; PD = progressive disease.

**Table 3 cancers-15-02450-t003:** Type of surgery.

**Conservative Breast Surgery (CBS)**	
**Quadrantectomy**	25 (49.0%)
**Oncoplastic surgery level II**	8 (15.8%)
**Conservative Mastectomy** - Nipple Sparing - Skin Sparing	18 (35.2%) - 9 (17.6%) - 9 (17.6%)
**Radical Breast Surgery (RBS)**	
**Modified radical mastectomy**	45 (100%)
**Axillary Surgery**	
Only Sentinel Lymph Node Biopsy Axillary Dissection	22 (22.9%) 73 (76.0%)

**Table 4 cancers-15-02450-t004:** Pathological characteristics.

Characteristics	All	CBS	RBS	*p*-Value
	96 Patients	51 (53.1%)	45 (46.9%)	
**BREAST**				
**pCR**	25 (26.0%)	16 (31.4%)	9 (20.0%)	*p = 0.248*
**ypT**				***p* < 0.0001**
- **0**	25 (26.0%)	16 (31.4%)	9 (20.0%)	
- **mic**	6 (6.3%)	4 (7.8%)	2 (4.4%)	
- **1a**	7 (7.3%)	6 (11.8%)	1 (2.2%)	
- **1b**	7 (7.3%)	6 (11.8%)	1 (2.2%)	
- **1c**	14 (14.6%)	9 (17.6%)	5 (11.1%)	
- **2**	16 (16.7%)	8 (15.7%)	8 (17.8%)	
- **3**	7 (7.3%)	2 (3.9%)	5 (11.1%)	
- **4**	14 (14.6%)	0 (0%)	14 (31.1%)	
**Multifocality**	36 (37.5%)	18 (35.3%)	18 (40.0%)	*p = 0.677*
**Pathological stage**				** *p < 0.0001* **
- **0**	19 (19.8%)	12 (23.5%)	7 (15.6%)	
- **I**	11 (11.5%)	9 (17.6%)	2 (4.4%)	
- **IIA**	21 (21.9%)	17 (33.4%)	4 (8.9%)	
- **IIB**	10 (10.4%)	5 (9.8%)	5 (11.1%)	
- **IIIA**	13 (13.5%)	5 (9.8%)	8 (17.8%)	
- **IIIB**	8 (8.3%)	0 (0%)	8 (17.8%)	
- **IIIC**	14 (14.6%)	3 (5.9%)	11 (24.4%)	
**Skin involvement**	13 (13.5%)	0 (0%)	13 (28.9%)	** *p < 0.0001* **
**Muscle involvement**	5 (5.2%)	0 (0%)	5 (11.1%)	** *p = 0.020* **
**AXILLA**				
**Sentinel node biopsy**	22 (22.9%)	16 (31.4%)	6 (13.3%)	*p = 0.051*
**Axillary dissection**	74 (77.1%)	35 (68.6%)	39 (86.7%)	*p = 0.051*
**Number of lymph nodes excised**	12.3 ± 7	11.4 ± 6.1	13.4 ± 7.8	*p = 0.146*
	(11; 7–16)	(10; 7–16)	(11; 8–15)	
**Number of metastatic lymph nodes**	3.9 ± 6.1	2 ± 4	6.1 ± 7.4	** *p = 0.001* **
	(1; 0–5)	(0.3; 0–2)	(4; 0.4–9)	
**ypN**				** *p = 0.004* **
- 0	31 (32.3%)	22 (43.1%)	9 (20.0%)	
- 1	33 (34.4%)	20 (39.2%)	13 (28.9%)	
- 2	18 (18.8%)	6 (11.8%)	12 (26.7%)	
- 3	14 (14.6%)	3 (5.9%)	11 (24.4%)	
**Positive lymph nodes/total lymph nodes excised ratio**	0.28 ± 0.3	0.15 ± 0.3	0.41 ± 0.4	** *p < 0.0001* **
	(0.11; 0–0.6)	(0.03; 0–0.2)	(0.33; 0.04–0.76)	

CBS = conservative breast surgery; RBS = radical breast surgery; pCR = pathological complete response.

**Table 5 cancers-15-02450-t005:** Univariate e multivariable analysis for conservative surgery.

Characteristics	Univariate Analysis			Multivariable Analysis		
	OR	*p* Value	95% CI	OR	*p* Value	95% CI
Menopausal status	1.249	*0.594*	0.551–2.833			
BRCA pathological mutations	4.783	*0.161*	0.537–42.582			
Istotype	0.871	*0.531*	0.565–1.342			
Grading	0.788	*0.573*	0.345–1.802			
cT4						
- a	21.098	*1.000*	0			
- b	**6.530**	** *<0.0001* **	**2.699–16.895**	**6.941**	** *0.021* **	**1.245–35.543**
- c	0.222	*0.07*	0.044–1.128			
- d	**0.168**	** *0.001* **	**0.068–0.489**	0.934	*0.941*	0.155–5.620
Tumor Subtype						
- Luminal	0.778	*0.54*	0.348–1.737			
- Triple negative	1.272	*0.642*	0.461–3.508			
- HER2 positive	1.125	*0.791*	0.469–2.699			
Median tumor size (mm)						
- <50	**2.491**	** *0.033* **	**1.079–5.753**	2.692	*0.095*	0.841–8.620
- 50.1–80	0.906	*0.819*	0.390–2.107			
- >80.1	**0.318**	** *0.026* **	**0.116–0.874**	0.936	*0.926*	0.228–3.837
cN	0.665	*0.073*	0.413–1.041			
Reduction of max size (%)						
- 100	0.538	*0.236*	0.194–1.498			
- 99.9–25.1	1.826	*0.146*	0.810–4.115			
- <25 or progression	**0.305**	** *0.01* **	**0.123–0.756**	**0.291**	** *0.042* **	**0.090–0.955**
ycN1	**0.331**	** *0.009* **	**0.144–0.761**	0.395	*0.078*	0.140–1.111

**Table 6 cancers-15-02450-t006:** Oncological outcomes according to the surgery performed.

	All 96 Patients	CBS 51 (53.1%)	RBS 45 (46.9%)	*p*-Value
**Locoregional Recurrence**				
** *N. of patients* **	19 (19.8%)	11 (21.6%)	8 (17.8%)	*p = 0.798*
Locoregional disease-free survival	74.9%	73.3%	77.4%	*LR = 0.801*
**Distant Recurrence**				
** *N. of patients* **	34 (35.4%)	11 (21.6%)	23 (51.5%)	** *p = 0.003* **
Distant disease-free survival	52.3%	64.8%	44.7%	** *LR = 0.002* **
**Death**				
** *N. of patients* **	29 (30.2%)	9 (17.6%)	20 (44.4%)	** *p = 0.007* **
Overall survival	60.2%	71.2%	48.6%	** *LR = 0.005* **

CBS = conservative breast surgery; RBS = radical breast surgery.

## Data Availability

The data presented in this study are available in this article.

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
