# Peer review of "Conservative Surgery in cT4 Breast Cancer: Single-Center Experience in the Neoadjuvant Setting"

_cancers, 2023, doi:10.3390/cancers15092450_

Round 1

Reviewer 1 Report

The article is an interesting one and addresses an often-difficult problem in the management of patients with advanced breast cancer, the decision of CBS versus RBS surgical therapy.

It is interesting that the authors present from the TNM staging evaluation data only the T criterion, very little about N and not at all about M, not to mention the rest of the staging evaluation criteria from the resection pieces: vascular invasion, perineural, etc. By consequence, are we referring only to patients with advanced local cancer without distant neoplastic seeds? Where do we therapeutically place patients with T4 classification but with distant metastases?

A second aspect is related to the lack of pathological prognostic stage, which appears as mandatory for a surgical decision. Based on this, the therapeutic decision is validated and, subsequently, confirmed by the resection pieces.

The third aspect. Taking into account the cosmetic aspect is important, but the aesthetic aspect is also important in patients who have not been "bothered" by this, so that they end up with tumors with skin ulcers, with skin, muscle, bone neoplastic invasions, often superinfected, hemorrhagic, etc? That is why the decision for CBS versus RBS filtered by cosmetic and not oncological considerations seems bizarre.

The fourth aspect refers to the data that evaluates the pathological response after NA. The strictly dimensional evaluation is insufficient to guide the patient towards a surgical therapy or another. For this, the clinical examination of the reconstruction possibilities depending on the lack of skin material is sufficient.

The fifth aspect is a truth of the Polichinele type. Those with incomplete or no response are obviously patients with aggressive forms of cancer, with tumor behavior that is difficult to control. The resection gesture has too little importance from a curative point of view, but only one of local sanitation, hemostatic, cytoreductive and this time also cosmetic. That is why it is not surprising that the long-term results are clearly unfavorable in these patients, and this is not due to the type of surgery but to the nature of the tumor, cancers with more or less aggressive potential.

The sixth element is a modest bibliography for an extremely complicated problem, which can be well filled out.

Author Response

Thank you for your comments, suggestions and very interesting considerations. We have tried to supplement the paper with the inputs you suggested. 

Reviewer 2 Report

This is a retrospective cohort study of all consecutive patients at one centre undergoing Neoadjuvant therapy for T4 breast cancer

Methods

- inclusion and exclusion clear

- abbreviations for which quadrant for surgery  need clarification "Q", "QSE/PA", "QSI,QIE"

- Follow up: 

   - was follow up via scheduled visit

   - (stadiation - check whether this is "staging")

- were any patients lost to follow up

Results:

- were there no Grade 1 tumours? I note that 7.3% of the cohort were Lumnal A, and 8% had low Ki67

- in those who had a CR by RECIST criteria, was the whole tumour bed planned to  removed?

- Table 5, Fig1, Fig3  - some decimal points are shown as commas

Conclusions:

- non CR patients do worse, CR patients have equivalent outcomes regardless of the type of surgery

- I note that Breast conservation was offered to 7/28 patients with inflammatory breast cancer - this would be worth discussing, especially if their outcome was no worse.

- was there a complication rate in those having immediate reconstruction?

- overall, the conclusions fit the data.

Figures  and Tables

- good

Author Response

Thank you for your comments, suggestions and reflections. We tried to supplement the text with the insights you suggested. 

Round 2

Reviewer 1 Report

Thanks to the authors for the changes and additions. Agree to publish.

Reviewer 2 Report

I accept the changes made by the authors